# Alzheimer's Disease blood biomarkers measured through remote capillary sampling correlate with cognition in older adults

Anne Corbett[1] ✉, Millie Sander-Long[1], Nicholas J. Ashton [2,3,4], Hanna Huber[2,5], Jakub Vavra[2], Luisa Sophie Braun-Wohlfahrt [2], Henrik Zetterberg [2,6,7,8,9,10,11], Laia Montoliu-Gaya [2], Jeffrey Cummings [12], Freya Bateman [1], Christine Davis[1] & Clive Ballard [1]

Blood biomarkers are rapidly becoming established for Alzheimer's Disease (AD) diagnosis. However, there is a need for more scalable tools to reach the 99% of individuals with early cognitive impairment who are not seen in specialist healthcare services. A recent study validated a capillary blood sampling technique to detect the p-tau217 and GFAP biomarkers. Here we used our PROTECT research study to show that these biomarkers, when collected using self-administered fingerprick tests, correlate well with venous blood biomarkers and with cognition and function in 174 people who were cognitively normal or who had mild cognitive impairment or AD. They can be used in combination with computerised cognitive testing to identify people with the highest risk of AD. The GFAP biomarker appears to be associated with vascular risk, unlike p-tau217. Patient feedback indicates high acceptability and usability of the capillary test method, giving confidence in the feasibility of this technology. The work suggests that capillary blood biomarkers could be used to enable triage of people with varying levels of risk of AD in clinical practice and for clinical trials, and could be used outside of clinical settings.

Blood biomarkers are increasingly used for dementia detection, diagnosis, and monitoring. They are central to verifying key Alzheimer's Disease (AD) pathologies as part of diagnostic criteria. The most robust blood biomarker is phosphorylated tau at amino acid 217 (p-tau217), which demonstrates high accuracy for the detection of AD pathology in validation studies against amyloid and tau PET[1–4], and is now FDA-approved for use in symptomatic patients undergoing investigation for cognitive complaints. These recent advances have

[1]University of Exeter Medical School, University of Exeter, Exeter, UK. [2]Department of Psychiatry and Neurochemistry, Institute of Neuroscience & Physiology, the Sahlgrenska Academy at the University of Gothenburg, Mölndal, Sweden. [3]Banner Alzheimer's Institute and University of Arizona, Phoenix, AZ, USA. [4]Banner Sun Health Research Institute, Sun City, AZ, USA. [5]German Center of Neurodegenerative Diseases, Bonn, Germany. [6]Clinical Neurochemistry Laboratory, Sahlgrenska University Hospital, Mölndal, Sweden. [7]Department of Pathology and Laboratory Medicine, University of Wisconsin School of Medicine and Public Health, Madison, WI, USA. [8]Wisconsin Alzheimer's Disease Research Center, University of Wisconsin School of Medicine and Public Health, University of Wisconsin-Madison, Madison, WI, USA. [9]Department of Neurodegenerative Disease, UCL Institute of Neurology, Queen Square, London, UK. [10]UK Dementia Research Institute at UCL, London, UK. [11]Centre for Brain Research, Indian Institute of Science, Bangalore, India. [12]Chambers-Grundy Center for Transformative Neuroscience, Department of Brain Health, Kirk Kerkorian School of Medicine, University of Nevada Las Vegas, Las Vegas, Nevada, USA. ✉e-mail: a.m.j.corbett@exeter.ac.uk

enabled the development of more refined concepts and diagnostic criteria, such as MCI due to AD, which are now seen as the gold standard for clinical trials and research[1–5]. Other blood biomarkers have been proposed to have a supportive role in diagnostic interpretation and potentially offer insights into disease pathophysiology. Glial fibrillary acidic protein (GFAP), a marker of astrogliosis and potentially neuroinflammation, has been shown to be associated with Aβ deposition, progression of MCI to AD dementia, progression of white matter hyperintensities, and with progression of cognitive decline in Parkinson's Disease[6–9].

Venous blood sampling is increasingly being adopted into the diagnostic pathway due to its affordability and feasibility compared with neuroimaging and CSF sampling. However, it still requires patients to visit specialist clinics. This is a considerable barrier to diagnosis and a major driver of screen failures in clinical trials since only 1 in every 1000 people with early cognitive decline currently access specialist healthcare in the UK[10]. This is also a global phenomenon particularly in rural areas and those with underdeveloped clinical infrastructure. There is therefore an urgent need for biomarker tests that can be used at scale outside of specialist settings. Such an approach would be transformational in enabling triaging of high-risk individuals, providing a new component to the diagnostic pipeline. It would also be an essential asset for supporting the efficient identification of eligible patients for the rapid growth of therapeutic trials in preclinical AD.

Self-administered microfluidic blood sampling technologies have now been developed for the measurement of AD blood biomarkers, including p-tau217 and GFAP, via at-home fingertip capillary blood collection on dried blood spot cards. The recent DROP-AD study reported in Nature Medicine demonstrated robust feasibility and validation of this method in 337 participants, showing significant correlations between capillary and venous blood p-tau217 (r = 0.79), and importantly also providing validation between capillary p-tau217 and

CSF Aβ42/pTau181 (r = 0.86). It also showed robust discrimination of individuals with a pre-determined Aβ-PET threshold (r = 0.868). The study showed good self-test validity with strong concordance between self-testing and nurse-led testing in the clinic.

In this study, we examined the potential clinical utility of this scalable biomarker technology as a self-administered capillary blood test completed at home and returned by post, and its potential value as a triage resource when used remotely in unsupervised settings. This study investigated the correlation of remotely collected blood p-tau217 and GFAP and cognitive biomarkers in 174 people who were cognitively normal or who had MCI or AD. It examines the potential value of the GFAP capillary biomarker to add value to p-tau217 as a triage tool to identify people at risk of AD and other causes of progressive cognitive impairment. We also confirmed the concurrent validity of the remotely collected samples against venous blood measurements and explored acceptability data from older adults to inform feasibility for future implementation as part of routine healthcare.

## Results

### Cohort Characterisation

This study recruited 226 patients, of whom 174 (77%) completed the blood test kits (146 normal cognition, 28 dementia) as part of the PROTECT programme in the UK, following the flow diagram in Fig. 1. The mean age of participants was 66.03 (SD 9.17), and 93 (54%) were female. p-tau217 data were available for 143 participants and GFAP data were available for all participants. Venous blood samples were available for 40 participants.

### Correlation between p-tau217 and GFAP with cognition and function

Analysis of cognitive data in the full cohort showed significant correlations between both capillary p-tau217 and GFAP with cognition. Capillary p-tau217 significantly correlated with episodic memory

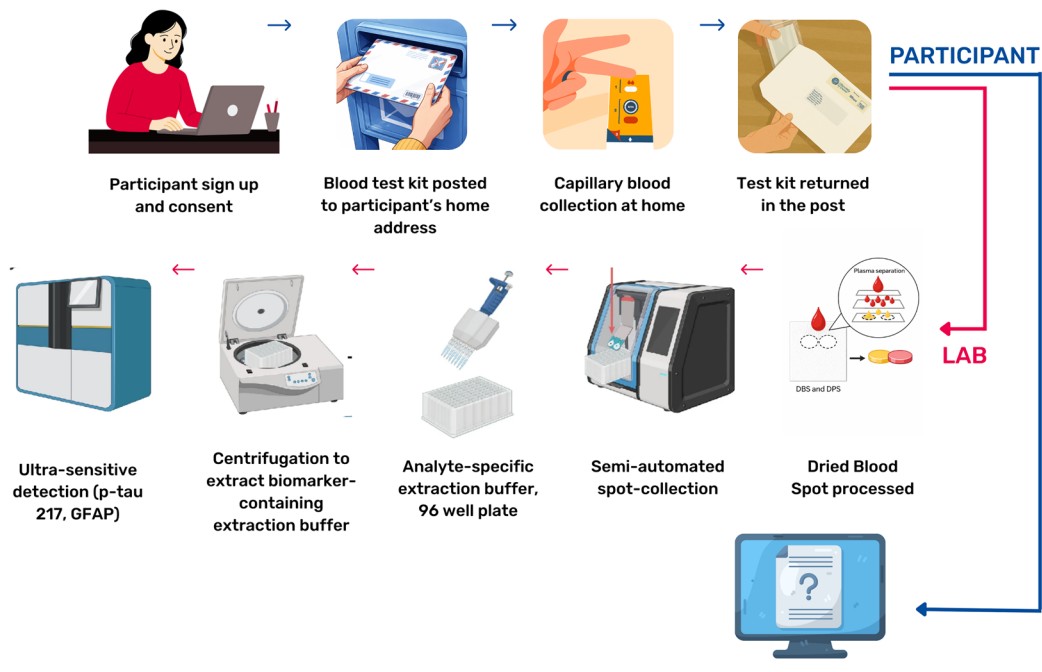

**Fig. 1 | Study flow diagram for sample collection and processing.** Collection and processing of capillary blood samples. A finger-prick sample was completed by participants at home without supervision, supported by instructional materials. Completed kits were returned by post. Blood spots were collected and incubated with analyte-specific extraction buffer, centrifuged and analysed using immunoassays on the Simoa platform to analyse p-tau217 and GFAP concentration. Image created in Canva under the available content license agreement (https://www.canva.com/policies/content-license-agreement/) and in BioRender.com (https://www.biorender.com/) under the available license agreement. Images also replicated with permission from Huber et al.[17].

**Table 1 | Performance of the p-tau217 and GFAP capillary biomarkers and pre-specified risk threshold in association with cognition and function (Spearman's Correlation)**

| Measure | | Correlation with capillary p-tau217 | | Correlation with capillary GFAP | |
|---|---|---|---|---|---|
| | | R | P | R | P |
| **Cognition** | Working Memory | 0.127 | 0.105 | 0.183 | 0.034 |
| | Episodic Memory | 0.299 | <0.001 | 0.169 | 0.068 |
| | Attention | 0.197 | 0.019 | 0.163 | 0.080 |
| | Executive Function | 0.191 | 0.021 | 0.182 | 0.046 |
| **Function** | IADL | 0.293 | <0.001 | 0.230 | 0.006 |
| | IQCODE | 0.244 | <0.001 | 0.098 | 0.243 |

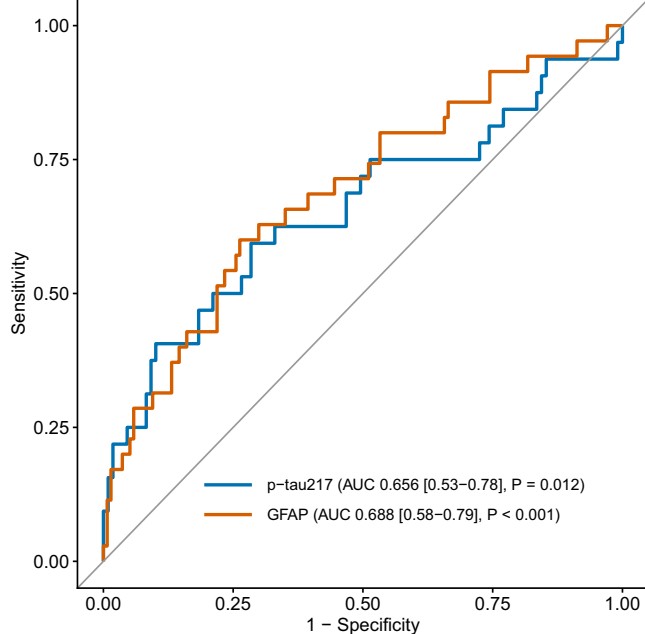

**Fig. 2 | Capillary biomarkers are sensitive to dementia status.** Receiver Operator Curve plots for discrimination of dementia by capillary p-tau217 (blue, n = 143) and GFAP (orange, n = 174). X axis shows 1-specificity to dementia status as confirmed by clinical diagnosis; Y axis shows sensitivity to dementia status. Grey reference line shows default performance of no discriminative ability. For numerical representation of the performance of the biomarkers, we present Area Under the Curve (AUC) alongside their *P* values and 95% Confidence Intervals (in square brackets). Source data are provided as a Source Data file.

(r = 0.299, P < 0.001), attention (r = 0.197, P = 0.019) and executive function (r = 0.191, P = 0.021). GFAP showed significant correlation with working memory (r = 0.183, P = 0.034) and executive function (r = 0.182, P = 0.046). Capillary p-tau217 also correlated with both functional measures of IADL (r = 0.293, P < 0.001) and IQCODE (r = 0.244, P < 0.001), and GFAP correlated with IADL (r = 0.230, P = 0.006) but not IQCODE (r = 0.098, P = 0.243) (Table 1). In a ROC analysis, capillary p-tau217 and GFAP achieved significant discrimination of individuals with and without dementia (p-tau217 AUC = 0.656, P = 0.012; GFAP AUC = 0.688, P < 0.001) (Fig. 2).

Although the ROC curve demonstrates a significant difference, this is not sufficient as a standalone diagnostic test. The goal here, however, is triage, not diagnosis. Therefore, a further analysis was undertaken to determine the differences in cognition above and below a pre-specified 85% specificity 35% sensitivity) threshold for each test (0.016 pg/ml for capillary p-tau217 and 12.45 pg/ml for capillary GFAP). These values identified 46 (26%) and 29 (17%) participants using the

p-tau217 and GFAP thresholds, respectively. ROC analysis showed that plasma p-tau and GFAP robustly discriminated between individuals falling above and below these thresholds defined by the capillary p-tau217 (AUC = 0.785, P < 0.001) and capillary GFAP (AUC = 0.895, P < 0.001) (Fig. 3).

Both biomarkers showed significant separation in both functional measures between individuals above and below the 85% specificity threshold, demonstrating significant differences with strong effect sizes for the majority of cognitive and functional measures (Table 2).

To further explore the potential utility of capillary p-tau217 as triage tool, a distribution plot of p-tau217 and memory performance illustrated a dual threshold approach to risk profiling, identifying a high-risk group requiring further diagnostic evaluation (above the 85% specificity threshold for capillary p-tau217 and meeting criteria for Age Associated Cognitive Decline for memory as defined by standardised protocols[11,12]) and a low-risk group (below the 85% specificity threshold and performing 1 SD or more above Standard Deviation for memory) (Fig. 4). Individuals in the high-risk quartile performed significantly worse in Attentional Speed, Accuracy and Executive Function, and both functional measures of IADL and IQCODE compared with individuals outside of this high-risk quadrant, and the low-risk individuals performed significantly better in all domains (Table 3).

### Relationship between capillary p-tau217 and GFAP
Even though both p-tau217 and GFAP are significantly associated with dementia and impaired cognition, in a comparison between individuals fulfilling the threshold criteria only 11 participants (6%) were positive for both capillary GFAP and p-tau217. The full breakdown of GFAP and p-tau217 status for the cohort are shown in Table 4.

To further understand potential differences in individuals identified with the two biomarkers a further analysis explored the relationship between biomarker status and cardiovascular risk factors (heart disease and hypertension). GFAP-positive individuals were nearly five times more likely to report a history of heart disease compared to GFAP-negative individuals, and there was a non-significant 69% increase in the likelihood of GFAP-positive individuals having a history of hypertension (Table 5). P-tau217 positivity had no significant impact on the likelihood of heart disease (P = 1.0) or hypertension (P = 0.13).

### Correlation between plasma and capillary blood biomarkers
In the subgroup of participants with venous blood samples (n = 40) correlation between venous and capillary p-tau217 and GFAP was significant and robust in patients with dementia (p-tau217: r = 0.711, P < 0.001; GFAP: r = 0.79, P < 0001) and without dementia (p-tau217: r = 0.743, P < 0.001; GFAP: r = 0.700, P < 0.001).

### Acceptability and feasibility for patients
Study participants provided feedback on the acceptability of the capillary blood test process. Their responses showed a high level of compliance and willingness to complete the tests at home, with 80% of participants completing the test without any need for help. 78% of participants reported that they would be willing to complete the tests remotely as part of their routine healthcare without supervision from a clinician (Fig. 5). Qualitative responses indicated an overall willingness, with dominant themes focusing on the ease and painlessness of the process, the potential for cost- and time-saving for patients and healthcare services, and for the potential benefit of monitoring health risks.

## Discussion
This paper highlights the potential clinical utility of the p-tau217 and GFAP capillary markers by demonstrating encouraging associations with cognitive and functional outcomes. The study also confirms the high and significant correlation between capillary and venous blood p-tau217 and GFAP, demonstrating similar correlations to those

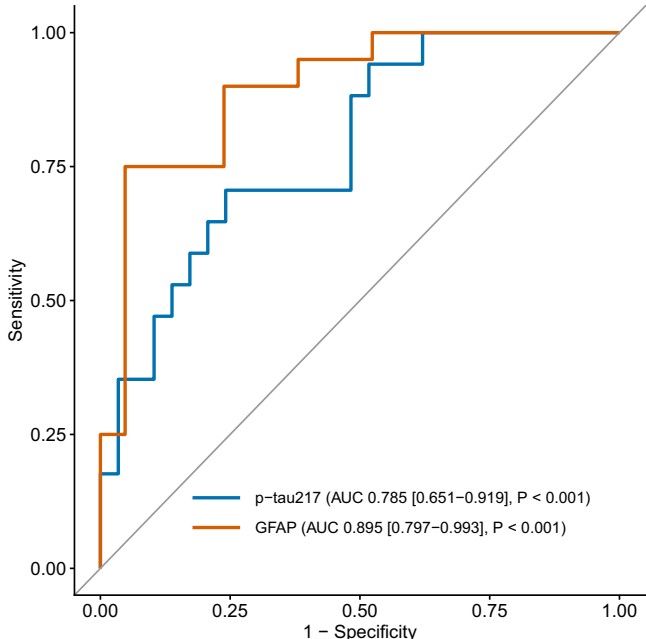

**Fig. 3 | Plasma biomarkers discriminate capillary cut-off risk groups.** Receiver Operator Curve plots for discrimination of individuals falling above and below the capillary p-tau217 85% specificity cut-off (0.016 pg.ml) by plasma p-tau217 (blue, n = 40) and for discrimination of individuals falling above and below the capillary GFAP 85% specificity cut-off (12.45 pg/ml) by plasma GFAP (orange, n = 40). The x-axis shows specificity for dementia, as confirmed by clinical diagnosis; the y-axis shows sensitivity for dementia status. The grey reference line shows the default performance, which is no discriminative ability. To numerically represent the performance of the biomarkers, we present the Area Under the Curve (AUC) alongside their P values and 95% Confidence Intervals (in square brackets). Source data are provided as a Source Data file.

**Table 2 | Difference in cognitive function in individuals above and below pre-defined 85% specificity thresholds for capillary p-tau217 and GFAP (Two-sided t-test)**

| Measure | | Difference between patients above vs below 85% specificity threshold | | | |
| --- | --- | --- | --- | --- | --- |
| | | Capillary p-tau217 | | Capillary GFAP | |
| | | Cohen's D | *P* | Cohen's D | *P* |
| **Cognition** | Memory | 0.548 | 0.020 | 0.465 | 0.037 |
| | Attention | 0.443 | 0.048 | 0.467 | 0.069 |
| | Executive Function | 0.339 | 0.137 | 0.541 | 0.016 |
| **Function** | IADL | 0.542 | 0.017 | 0.754 | 0.004 |
| | IQCODE | 0.657 | 0.001 | 0.6540.230 | 0.012 |

recently reported in the DROP-AD study against venous blood and CSF markers, but extending this concurrent validity to capillary blood samples that were self-collected remotely and returned by post. Furthermore, the findings provide greater confidence in the acceptability and feasibility of the technology for patients to use remotely at home as a self-completed test returned by post.

Capillary p-tau217 was significantly higher in people with dementia compared to those without, and was significantly associated with cognitive performance and function. The ROC analysis shows results that are consistent with differentiation between a healthy and a dementia cohort, thus providing confidence in this approach. The 85% specificity threshold for p-tau217, which aligned with the optimal threshold from the DROP-AD study and would be the recommended threshold for triage, showed an even stronger discrimination for the identification of individuals with cognitive and functional impairments. The scatter plot and dual thresholding further supported this recommendation and suggests that the combination of p-tau217 and computerized neuropsychology has the potential to be used to identify a high-risk group who would potentially benefit from a more detailed diagnostic assessment, and a low-risk group that would not require any further support. It also enables identification of an intermediate-risk group that would benefit from ongoing monitoring.

This plot suggested that a combination of the capillary p-tau217 85% specificity threshold, and episodic memory performance 1 SD below benchmarked norms was a pragmatic means of identifying a potentially high-risk group of 9% of participants who also showed significantly higher impairment in cognition and function. Importantly, this threshold for impairment of episodic memory indicates a much milder level of impairment than the 1.5 SD change required as part of identifying people with Mild Cognitive Impairment (MCI), but

in combination with elevated capillary p-tau 217 has the potential to triage a group at a milder stage of pre-clinical impairment. The dual thresholding approach to identifying high and low risk groups would enable more sensitive application of this technology to triage individuals with high priority risk, those requiring onward monitoring, and those who do not. Further work would be needed to better characterise the intermediate group and establish the most meaningful approach for onward monitoring in the community and primary care.

This work raises the potential for the provision of a remote triage method for identifying individuals likely to be biomarker positive for AD and with a likelihood of impairments in cognition and function by a simple at-home test. With further validation, such a technology could improve the clinical pathway for the triage and assessment of people with pre-clinical AD. Currently, only one in 1000 people with early cognitive impairment receives a specialist evaluation. The combination of a capillary test for p-tau217 and computerized neuropsychology offers a triage approach which could potentially provide a straightforward, efficient and cost-effective method to reach large numbers of individuals in the community who would not otherwise receive diagnosis or support and to optimise the clinical pathway to enable early detection of the highest risk individuals. It is important to acknowledge that the potential utility for this technology would be for triage, not diagnosis, particularly due to the low sensitivity of the risk threshold parameters. There are also potential opportunities to utilize this approach as a pre-screening triage step to improve the efficiency of clinical trials focussing on disease-modifying therapies for pre-clinical AD, where screen failure rates often reach 90%[13]. Ongoing light touch virtual monitoring would be recommended for individuals in the intermediate risk groups which could be achieved through at-home monitoring with computerised neuropsychology, as has been validated by the PROTECT programme over the last decade.

The study also examined the potential value of GFAP as an additional biomarker screen. Unexpectedly, even though both ptau217 and GFAP both identified individuals with cognitive impairment, there was only a modest overlap in individuals who were positive for both GFAP and p-tau217, with GFAP identifying a different group of at-risk individuals. Further analysis demonstrated an association between GFAP positivity and heart disease and a differential pattern of cognitive impairments with greater attentional impairments in participants with elevated GFAP. These observations support the hypothesis that GFAP may be identifying individuals at risk of cognitive impairment related to vascular factors and with non-AD causes of neuroinflammation[14]. In clinical practice where clinicians are trying to identify people at risk of all cause progressive cognitive impairment, GFAP may therefore be a useful additional biomarker. However, for clinical trials focussing on pre-clinical AD, p-tau217 alone is likely to be the preferred triage solution.

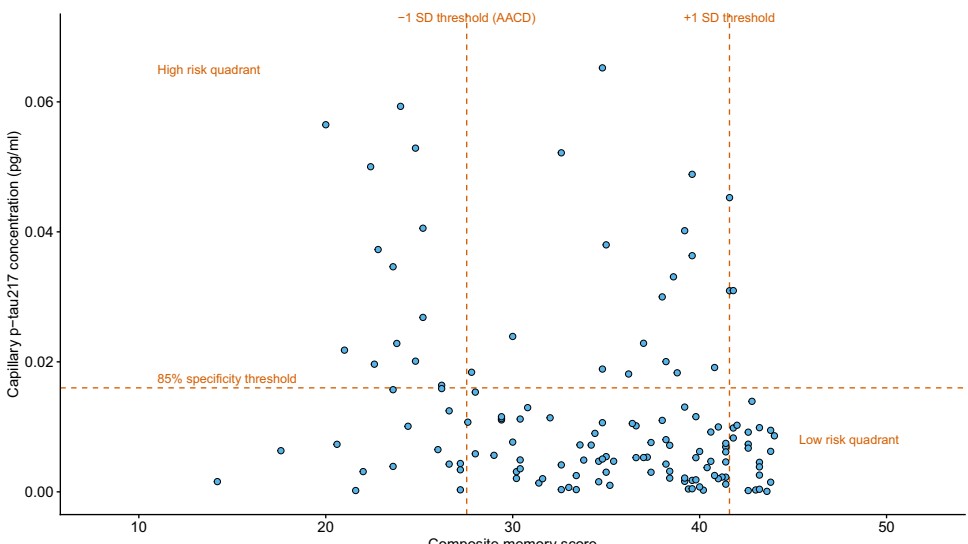

**Fig. 4 | Dual threshold approach identifies high, medium and low risk groups based on capillary p-tau217 and memory score.** Scatter plot of capillary p-tau217 and performance on the composite memory score on the PROTECT Cognitive Test System. Dual threshold cut-offs are 85% specificity threshold for capillary p-tau217

(0.016 pg/ml) and for +/- one standard deviation (SD) from the mean for Age-Associated Cognitive Decline (−1SD) and superior performance (+1 SD). High and low risk quadrants are identified based on individuals falling above and below the dual-threshold cutoffs. Source data are provided as a Source Data file.

**Table 3 | Individuals in the highest and lowest capillary risk p-tau217 and memory profile show significantly different cognition and function compared with the rest of the cohort (Two-sided t-test)**

| Measure | | High Risk Quartile (n = 15) | | Low Risk Quartile (n = 27) | |
|---|---|---|---|---|---|
| | | Cohen's D | P | Cohen's D | P |
| **Cognition** | Speed of Attention | 1.04 | 0.004 | 0.59 | <0.001 |
| | Attentional Accuracy | 0.97 | 0.007 | 0.62 | <0.001 |
| | Executive Function | 1.32 | <0.001 | 1.47 | <0.001 |
| **Function** | IADL | 1.43 | <0.001 | 0.69 | <0.001 |
| | IQCODE | 1.15 | <0.001 | 0.48 | <0.001 |

This study demonstrates significant strengths. The use of unsupervised capillary testing is new to the field, and these findings provide clear evidence regarding their validity outside of clinical settings in a cohort of both healthy and cognitively impaired individuals. The qualitative findings regarding the acceptability of the capillary tests further enhance the rigor of this work, providing confidence in its translatability to real-world settings. There are, however, some important limitations to acknowledge. Whilst 80% of participants were able to successfully complete the self-testing with a combination of PPI developed video and written instructions, it will be important to further improve these materials and provide additional virtual support to enable higher completion levels. In addition, although there was a good level of correlation (>0.7) between capillary and venous p-tau217 there may be opportunities to streamline the collection and processing protocols to further improve the concurrent validity. In the GFAP analysis cardiovascular risk factors were captured using a single self-report item. This could be expanded with clinical data for increased accuracy. Further work is needed to replicate and expand this work, to validate these findings in an independent cohort, determine the longitudinal predictive value of the capillary biomarkers, validate against PET imaging with amyloid and tau ligands, and refine the methodology to incrementally improve sensitivity further. It will also

be important to establish a triage and monitoring process that interfaces with primary care and integrates into clinical and clinical trial pathways. Specifically, it will be important to examine its use alone or in combination with computerised neuropsychology as a triage tool in the community and primary care and as a valuable approach to improve clinical trial recruitment.

## Methods

### Ethical approval and governance
This study was conducted in accordance with the principles of the Declaration of Helsinki, Good Clinical Practice guidelines and all applicable regulatory requirements. Study protocol, participants' information sheets, and informed consent process and documents were approved by independent National Health Service (NHS) Research Ethics Committees (NHS REC London Bridge, Ref 13/LO/1578; NHS REC Greater Manchester East, Ref: 24/NW/0151). Study investigators obtained consent, recruited participants, collected data and adhered to ethical standards. The sponsor conducted centralised monitoring with strict ethical oversight.

### Recruitment
Participants with dementia were recruited through primary and secondary care sites across South-West UK and fulfilled criteria for mild to moderate dementia based on the National Institute of Neurological and Communicative Disorders and Stroke and Alzheimer's Disease and Related Disorders Association (NINCDS-ADRDA) criteria, with impairment in at least two cognitive domains with impact on functionality. Cognitively healthy participants were recruited through the nationwide PROTECT-UK ageing cohort using established online and digital protocols. All participants provided self-reported demographic information at the point of recruitment. Participants were not compensated for their involvement. The full data collection process is shown in the flow diagram in Fig. 1.

### Blood biomarker collection
All participants completed a capillary blood collection kit either at home or in an unsupervised setting. Blood was collected using the Capitainer® B50 and SEP10 (Capitainer AB, Solna, Sweden) dried blood spot collection device. Participants were instructed to obtain blood from the middle or index finger using a single-use lancet, following

**Table 4 | GFAP and -tau217 status of participants with available data for both biomarkers**

| Biomarker status | % Participants |
|---|---|
| GFAP Positive + p-tau217 positive | 8.3 |
| GFAP positive + p-tau217 negative | 11.9 |
| GFAP negative + p-tau217 positive | 16.7 |
| GFAP negative + p-tau217 negative | 62.9 |

**Table 5 | The association between GFAP biomarker status and cardiovascular risk factors (Logistic regression, adjusted for age and sex)**

| | Percent of positive cases | | Odds ratio (95% CI) | p value |
|---|---|---|---|---|
| | GFAP –ve | GFAP +ve | | |
| **Hypertension** | 24.6% | 35.7% | 1.32 (0.53 – 3.33) | 0.55 |
| **Heart disease** | 6.5% | 25.0% | **4.14 (1.31 – 13.10)** | **0.016** |

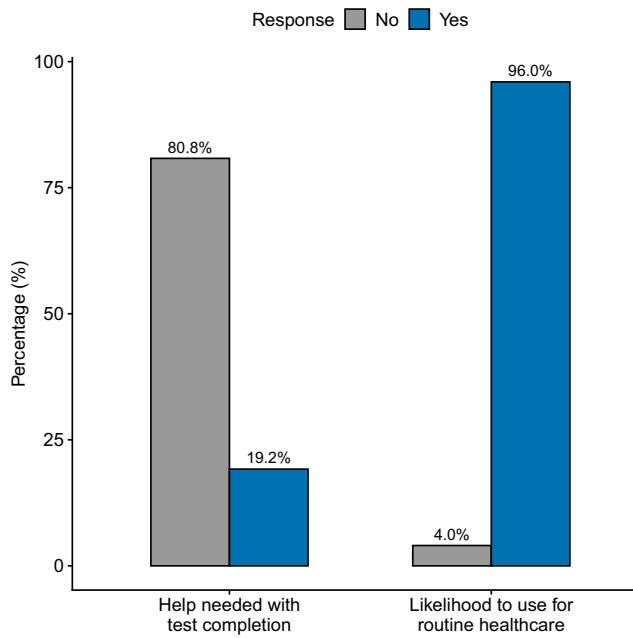

**Fig. 5 | Capillary blood sampling is acceptable and feasible for patients.** Participant feedback on the feasibility and acceptability of the capillary blood test for at-home use. Responses are shown as % Yes (blue) / No (grey) response to two questions, one asking whether users needed help with completing the capillary blood test, and one asking whether they would be likely to use these tests as part of routine healthcare. Source data are provided as a Source Data file.

video and written instructions. 70 ul of capillary blood were applied to the device and left to dry, returning a plasma-like sample. After drying, cards were stored at room temperature and shipped without temperature control or cooling to the central research team at the University of Exeter, and then batched and sent regularly to the Neurochemistry Laboratory, Gothenburg, Sweden. Cards were processed and extracted using standard protocols (Quanterix, Billerica, MA, USA). A subset of 40 participants provided a venous blood sample by venipuncture for biomarker analysis at a clinic visit, taken by a trained staff member. Samples were vortexed for 30 seconds at 2000 rpm and centrifuged at 4000 g and 20 °C for 10 minutes and batch-analysed at the end of the study. Biomarker analysis was completed by Single molecule array (Simoa) technology using a neat protocol

(ALZpath Simoa® GFAP Assay) (Quanterix, Billerica, MA, USA) to detect calibrated levels of p-tau217 and GFAP.

## Computerised cognitive assessment

All participants completed the PROTECT Cognitive Test System on a computer or touchscreen device. The system consists of tests for memory (Delayed Picture Recognition, Paired Associate Learning, Self-Ordered Search, Digit Span), executive function (Verbal Reasoning) and attention (Digit Vigilance, Simple and Choice Reaction Time) which can be combined into four composite outcomes for memory, attentional accuracy, speed of attention and executive function. Description of these tests and their outputs have been published previously[15,16]. Participants completed a brief practice session of each test in sequence prior to completing a full test session in order to reduce practice effects due to orientation to the system.

## Health and self-report data

All participants completed the Lawton Instrumental Activities of Daily Living Scale (IADL) and IQCODE scale as self-report measures. They also provided self-report data on cardiovascular risk factors of hypertension and heart disease through a medical history questionnaire on the PROTECT platform. Participants also completed a brief survey online to capture their experience of at-home use of the capillary blood sampling kits, using Likert scale responses to provide information on ease of use, acceptability and willingness to use the kits as part of routine healthcare and monitoring. Where appropriate, participants were supported in completing these measures by a caregiver.

## Statistical Analysis

No data were excluded from analyses. Correlation analyses were performed using Spearman's R to evaluate the correlation between venous and capillary biomarkers, and between cognitive and functional outcomes and the capillary p-tau217 and GFAP biomarkers. Receiver Operator Characteristic analyses were undertaken to define 85% specificity threshold for each biomarker, to determine discrimination of sub-threshold individuals using plasma biomarkers, and to examine discrimination of dementia by capillary biomarkers. Independent t-test analysis compared cognitive performance in individuals above and below the 85% specificity thresholds and between individuals defined as high and low risk based on p-tau217 and memory, compared with the rest of the cohort. Age-Associated Cognitive Decline and cognitively healthy thresholds were defined based on performing one Standard Deviation below and above age-matched norms in individual cognitive domains, based on published criteria[11,12]. Logistic regression was used to assess the relationship between GFAP positivity and cardiovascular risk factors, adjusting for age and sex. The outcome variable was binary, and all observations were independent. Influential data points were identified using Cook's distance and downweighted via inverse weighting. Multicollinearity was assessed using variance inflation factors (VIF < 5). Model robustness was evaluated by comparing coefficients, predicted probabilities, and AUC between original and weighted models. Model fit and influence were further assessed using deviance residuals. Odds ratios (ORs) and 95% confidence intervals (CIs) were calculated from model coefficients. Sample size was calculated using a power calculation for correlation of R = 0.7 between plasma and capillary tau, based on the DROP-AD study. All analyses were conducted using IBM SPSS Statistics (Version 29) and R statistical software (Version 4.4.3). All analyses were successfully reproduced using the same methodology.

## Reporting summary

Further information on research design is available in the Nature Portfolio Reporting Summary linked to this article.

## Data availability

The raw data are available under restricted access to protect participant identity. An anonymised minimum dataset will be made available for purposes of replication. Access can be obtained on request to the PROTECT Research Group (protect.data@exeter.ac.uk) and are subject to confirmation of a data usage agreement. Source data are provided with this paper.

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

## Acknowledgements

This study was funded by the NIHR Invention for Innovation funding stream (AC/CB; NIHR204824) and by the Exeter Biomedical Research Centre (AC/CB; NIHR203320). This paper represents independent research part-funded by the National Institute of Health Research (NIHR) Exeter Biomedical Research Centre and NIHR HealthTech Research Centre in Brain Health. The views expressed are those of the authors and not necessarily those of the NIHR or the Department of Health and Social Care. This paper was also supported by the NIHR Collaboration for Leadership in Applied Health Research and Care South-West Peninsula. HZ is a Wallenberg Scholar and a Distinguished Professor at the Swedish Research Council supported by grants from the Swedish Research Council (#2023-00356, #2022-01018 and #2019-02397), the European Union's Horizon Europe research and innovation programme under grant agreement No 101053962, Swedish State Support for Clinical Research (#ALFGBG-71320), the Alzheimer Drug Discovery Foundation (ADDF), USA (#201809-2016862), the AD Strategic Fund and the Alzheimer's Association (#ADSF-21-831376-C, #ADSF-21-831381-C, #ADSF-21-831377-C, and #ADSF-24-1284328-C), the European Partnership on Metrology, co-financed from the European Union's Horizon Europe Research and Innovation Programme and by the Participating States (NEuroBioStand, #22HLT07), the Bluefield Project, Cure Alzheimer's Fund, the Olav Thon Foundation, the Erling-Persson Family Foundation, Familjen Rönströms Stiftelse, Familjen Beiglers Stiftelse, Stiftelsen för Gamla Tjänarinnor, Hjärnfonden, Sweden (#FO2022-0270), the European Union's Horizon 2020 research and innovation programme under the Marie Skłodowska-Curie grant agreement No 860197 (MIRIADE), the European Union Joint Programme – Neurodegenerative Disease Research (JPND2021-00694), the National Institute for Health and Care Research University College London Hospitals Biomedical Research Centre, the UK Dementia Research Institute at UCL (UKDRI-1003), and an anonymous donor. Cognitive tests were provided by Cognitron, Wesnes Cognition (CogTrack™) and ECog Pro Ltd.

## Author contributions

A.C.: conceptualisation, data collection, data analysis and interpretation, manuscript writing and editing. M.S.L.: conceptualisation, data analysis and interpretation, manuscript writing and editing. C.B.: conceptualisation, data interpretation, manuscript writing, and editing. N.A.: data collection, data interpretation, manuscript editing and review. H.H.: data collection, data interpretation, manuscript editing, and review. J.V.: data collection, data interpretation, manuscript editing and review. L.S.B.W.: data collection, data interpretation, manuscript editing, and review. H.Z.: manuscript editing and review. L.M.: data collection, data interpretation, manuscript editing, and review. J.C.: manuscript editing and review. FB: study delivery, data collection, manuscript editing, and review. C.D.: study delivery, data collection, manuscript editing, and review

## Competing interests

Anne Corbett declares consultancy work for Novartis, Addex, Suven, Sunovion, Janssen, and Acadia pharmaceutical companies and grant funding from Novo Nordisk, ReMynd, Therini Bio pharmaceutical companies. Nicholas Ashton has received consultancy/speaker fees from Alamar Biosciences, Bioartic, Biogen, Eli-Lilly, Neurogen Biomarking, Roche, Spear Bio, Quanterix, and Vigil Neurosciences. Laia Montoliu-Gaya has received consultancy/speaker fees from Quanterix and Esteve. Clive Ballard has received consulting fees from Acadia Pharmaceutical Company, AARP, Addex Pharmaceutical Company, Eli Lily, Enterin Pharmaceutical Company, GWPharm, H. Lundbeck Pharmaceutical Company, Novartis Pharmaceutical Company, Janssen Pharmaceuticals, Johnson & Johnson Pharmaceutical, Novo Nordisk Pharmaceutical Company, Orion Corp Pharmaceutical Company, Otsuka America Pharm Inc, Sunovion Pharm. Inc, Suven Pharmaceutical Company,

Roche Pharmaceutical Company, Biogen Pharmaceutical Company, Synexus Clinical Research Organization, and tauX Pharmaceutical Company, and research funding from Synexus Clinical Research Organization, Roche Pharmaceutical Company, Novo Nordisk Pharmaceutical Company, and Novartis Pharmaceutical Company. Jeffrey Cummings has provided consultation to Acadia, Acumen, ALZpath, AnnovisBio, Artery, Axsome, Biogen, Bristol-Myers Squibb, Eisai, Fosun, GAP Foundation, Hummingbird Diagnostics, IGC, Janssen, Julius Clinical, Kinoxis, Lilly, LSP/eqt, Merck, MoCA Cognition, Novo Nordisk, NSC Therapeutics, Otsuka, ReMYND, Roche, Scottish Brain Sciences, Signant Health, Simcere, Sinaptica, and T-Neuro pharmaceutical, assessment, and investment companies. He is supported by NIGMS grant P20GM109025; NIA R35AG71476; NIA R25AG083721; NINDS RO1NS139383; Alzheimer's Disease Drug Discovery Foundation (ADDF); Ted and Maria Quirk Endowment; Joy Chambers-Grundy Endowment. Henrik Zetterberg has served at scientific advisory boards and/or as a consultant for Abbvie, Acumen, Alector, Alzinova, ALZpath, Amylyx, Annexon, Apellis, Artery Therapeutics, AZTherapies, Cognito Therapeutics, CogRx, Denali, Eisai, Enigma, LabCorp, Merck Sharp & Dohme, Merry Life, Nervgen, Novo Nordisk, Optoceutics, Passage Bio, Pinteon Therapeutics, Prothena, Quanterix, Red Abbey Labs, reMYND, Roche, Samumed, ScandiBio Therapeutics AB, Siemens Healthineers, Triplet Therapeutics, and Wave, has given lectures sponsored by Alzecure, BioArctic, Biogen, Cellectricon, Fujirebio, LabCorp, Lilly, Novo Nordisk, Oy Medix Biochemica AB, Roche, and WebMD, is a co-founder of Brain Biomarker Solutions in Gothenburg AB (BBS), which is a part of the GU Ventures Incubator Program, and is a shareholder of CERimmune Therapeutics (outside submitted work). All other authors declare no competing interests.
