## [Transparent Peer Review file · Nature Communications]

Alzheimer's Disease blood biomarkers measured through remote capillary sampling correlate with cognition in older adults

Corresponding Author: Professor Anne Corbett

Version 0:

Reviewer comments:

Reviewer #1

(Remarks to the Author)

The authors have addressed a number of this reviewer's prior comments, but several concerns remain.

1. The idea that capillary ptau217 and GFAP could be used together with computerized cognitive testing as a "triage" tool is certainly interesting and could have a use case, particularly for clinical trials. However, it is a little counterintuitive given the way screening tools are typically applied as highly sensitive (rather than specific) instruments that can indicate the need for follow up with more specialized testing. For example, the plasma ptau217 tests are currently used in such a way where individuals with clearly positive or negative results may not require further testing, and those with ambiguous levels between the two thresholds are sent for further testing with gold standard amyloid PET or CSF evaluations. In this case we are identifying those at highest risk but are likely missing many cases given the low sensitivity. Did the authors consider a dual threshold approach?
2. There is inconsistency in the description of the sample, which causes some confusion for the reader. Initially the sample is described as a group of cognitively normal participants and a group with "cognitive impairment". Later a group with "dementia" is mentioned. The methods section states that participants were diagnosed with AD dementia using NINCDS-ADRDA criteria. If the methods are accurate, this should be stated more clearly and consistently in the main text so the reader understands that we are discussing patients with a clinical diagnosis of AD dementia.
3. The ROC curves identifying those with dementia versus normal cognition are interesting, but one would not really expect them to be impressive since these patients only have a clinical diagnosis of AD dementia and may not be amyloid positive.
4. The authors identified a subgroup of the normal sample with "Age Associated Cognitive Decline". In the statistical analysis section, it is explained that this was defined as greater than 1 SD below normal performance in "individual cognitive domains". No citations for any criteria are provided. The concept of Age Associated Cognitive Decline is unclear, and given the analysis is cross-sectional there is no evidence of "decline" but rather performance on the low end of the distribution which could be premorbid. It is unclear whether the diagnosis is based on one or multiple tests or whether there are validated criteria for this identification. There are criteria for subtle cognitive dysfunction that may represent a Pre-MCI group of interest and would be more appropriate for this analysis, or simply analyzing those with lower memory performance without any diagnostic label as is done in the correlational analysis.
5. The statistical analysis of associations with cardiovascular disease does not seem to include any covariates. Cardiovascular disease often differs by age and sex, which would be important to take into account. It is also unclear how an independent samples t-test could test the relationship between ptau217 and the presence of cardiovascular disease. Could this have been a chi-square? Or perhaps the cardiovascular risk factors were counted?
6. It is unclear how cardiovascular disease was defined in the reporting scale (i.e., what were the risk factors studied and how were they defined?). The main text indicates "cardiovascular disease", the table indicates "heart disease", and the methods indicates "cardiovascular risk factors".
7. There is an overinterpretation of the associations with cardiovascular disease and some other findings given the study limits, which predominantly include small sample size and very limited participant characterization.

Version 1:

Reviewer comments:

Reviewer #1

(Remarks to the Author)

The authors have adequately addressed my comments.

NCOMMS-26-003218-T: Response to Reviewers

We would like to thank the reviewer for their further review of our manuscript. We have responded to their comments below and in the resubmitted manuscript

Comment 1: The idea that capillary ptau217 and GFAP could be used together with computerized cognitive testing as a “triage” tool is certainly interesting and could have a use case, particularly for clinical trials. However, it is a little counterintuitive given the way screening tools are typically applied as highly sensitive (rather than specific) instruments that can indicate the need for follow up with more specialized testing. For example, the plasma ptau217 tests are currently used in such a way where individuals with clearly positive or negative results may not require further testing, and those with ambiguous levels between the two thresholds are sent for further testing with gold standard amyloid PET or CSF evaluations. In this case we are identifying those at highest risk but are likely missing many cases given the low sensitivity. Did the authors consider a dual threshold approach?

Response: We would like to thank the reviewer for this helpful and insightful suggestion. We have implemented a dual-threshold approach to the methodology using the same memory cut-off approach. This enables identification of high and low-risk quadrants, with thresholds identifying individuals performing significantly worse and better in all cognitive and functional domains respectively. This approach also identifies individuals in an intermediate risk group who represent those who would potentially benefit from ongoing monitoring. This group will require further characterisation but are of particular interest from a clinical standpoint. This addition has been included in the methods, results and discussion.

Comment 2. There is inconsistency in the description of the sample, which causes some confusion for the reader. Initially the sample is described as a group of cognitively normal participants and a group with “cognitive impairment”. Later a group with “dementia” is mentioned. The methods section states that participants were diagnosed with AD dementia using NINCDS-ADRDA criteria. If the methods are accurate, this should be stated more clearly and consistently in the main text so the reader understands that we are discussing patients with a clinical diagnosis of AD dementia.

Response: Apologies for the lack of clarity. There are two groups – one cognitively normal cohort and one dementia cohort diagnosed using the NINCDS-ADRDA criteria. This has been clarified in the text.

Comment 3: The ROC curves identifying those with dementia versus normal cognition are interesting, but one would not really expect them to be impressive since these patients only have a clinical diagnosis of AD dementia and may not be amyloid positive.

Response: Thank you, we agree with the reviewer on this point. The ROC curves are consistent with the clinical status of the group. We have noted this in the text.

Comment 4: The authors identified a subgroup of the normal sample with “Age Associated Cognitive Decline”. In the statistical analysis section, it is explained that this was defined as greater than 1 SD below normal performance in “individual cognitive domains”. No citations for any criteria are provided. The concept of Age Associated Cognitive Decline is unclear, and given the analysis is cross-sectional there is no evidence of “decline” but rather performance on the low end of the distribution which could be premorbid. It is unclear whether the diagnosis

is based on one or multiple tests or whether there are validated criteria for this identification. There are criteria for subtle cognitive dysfunction that may represent a Pre-MCI group of interest and would be more appropriate for this analysis, or simply analyzing those with lower memory performance without any diagnostic label as is done in the correlational analysis.

Response: Apologies for not providing the context and reference for the Age-Associated Cognitive Decline definition. This is based on the published FDA criteria for cognitive impairment staging, and applied to the cognitive data presented in this paper. We have clarified this in the text and included a reference. We applied the AACD criteria in this study to fulfil our aim of identifying individuals with earlier, pre-clinical cognitive impairment as opposed to MCI or mild dementia, which would already be identified in clinics.

Comment 5. The statistical analysis of associations with cardiovascular disease does not seem to include any covariates. Cardiovascular disease often differs by age and sex, which would be important to take into account. It is also unclear how an independent samples t-test could test the relationship between ptau217 and the presence of cardiovascular disease. Could this have been a chi-square? Or perhaps the cardiovascular risk factors were counted?

Response: Thank you. The original analysis took a count of cardiovascular risk factors but we have taken a different approach to allow us to control for age and sex. We have replaced the previous analysis with a logistic regression. This does not change the outcome significance. The updated methods and results are included in the manuscript.

Comment 6: It is unclear how cardiovascular disease was defined in the reporting scale (i.e., what were the risk factors studied and how were they defined?). The main text indicates "cardiovascular disease", the table indicates "heart disease", and the methods indicates "cardiovascular risk factors".

Response: Apologies for the lack of clarity here. Participants were asked to report on medical conditions as part of a self-report questionnaire. This included the known cardiovascular risk factors, heart disease and hypertension. These terms have been aligned in the text.

Comment 7: There is an overinterpretation of the associations with cardiovascular disease and some other findings given the study limits, which predominantly include small sample size and very limited participant characterization.

Response: Thank you, we have softened the language around the implications of this work in the discussion.